# Co-Creating Strategies to Optimize Traditional Silvopastoral Systems through the Management of Native Trees in Caívas in Southern Brazil

Ana Lúcia Hanisch [1,*] and Lígia Carolina Alcântara Pinotti [2]

[1] Researcher EPAGRI—Canoinhas Experimental Station, Canoinhas 89466-500, Brazil
[2] Post-Graduate Program in Agronomy, Federal University of Paraná, Curitiba 80210-170, Brazil; pinotti@ufpr.br
[*] Correspondence: analucia@epagri.sc.gov.br

**Abstract:** The conservation of forest remnants in southern Brazil is closely related to historical land use, for example in systems such as *caívas* that occur within remnants of the Araucaria Forest and include livestock production and the extraction of yerba mate. Over the last decade, technologies adapted for these systems have been developed that promote a significant increase in animal productivity, without harming forest regeneration or the maintenance of the tree layer. However, the fertilization of pastures proposed in the technology has also promoted greater growth of native trees, with a consequent increase in shade levels. This, in turn, has affected the maintenance of pasture and yerba mate in the understory. Thus, this study sought to develop a methodology to adjust shade levels based on forest management that adheres to the limits permitted by current legislation. The objective was to evaluate the effect of tree management to maintain 50% shade levels on environmental indicators in a *caíva* that has been implementing pasture improvement technology since 2013. Native tree management occurred in 2020 and 2022 and the results were compared with data from the floristic survey of the area carried out in 2013. The results indicate that although the adoption of forest management to adjust shade levels reduced the density of individuals, it did not affect forest diversity, nor the basal area of the *caíva* tree layer. As such, it is possible to maintain pasture and yerba mate production in the area. Strategies like this are fundamental so that the forest landscape can continue to offer a source of production while also supporting environmental conservation.

**Keywords:** agroforestry systems; silvopastoral systems; *Ilex paraguariensis*; pasture; Araucaria Forest

## 1. Introduction

Silvopastoral agroforestry (SAF) systems are characterized by the intentional combination of grazing animals (cattle, sheep, goats, among others) with trees and forage in the same production unit at the same time [1]. The occurrence of Silvopastoral Systems (SSP) with native trees can be found in the most diverse environments around the planet, from savannas in Africa, *dehesas* in Spain and cork oak forests in Portugal, to traditional silvopastoral systems in Central America [1–3]. In Southern Brazil, *caíva* is a local term given to a more than 100-year-old SAF system that occurs within Araucaria Forest remnants, where the raising of dairy and beef cattle, the extraction of yerba mate (*Ilex paraguariensis* St. Hil.), and the maintenance of native tree species occur simultaneously [2,4–6].

Most *caívas* are currently found in forest fragments of varying sizes, which occupy around 130,000 hectares [2]. Despite being productive systems, with an almost constant presence of cattle, *caívas* contribute to maintaining significant forest cover in the region, including rare tree species and even some species threatened with extinction. Surveys conducted in *caívas* have confirmed high levels of tree species richness (an average of 40 species per hectare), with a density ranging from 220 to 1300 adult trees·ha$^{-1}$ [7–11], which confirms the importance of this traditional silvopastoral system for forest conservation.

Despite their unquestionable cultural, historical, and environmental importance, *caívas* are not economically attractive due to their very low animal productivity, with an average stocking rate of 0.35 cow·ha$^{-1}$. This low productivity is a consequence of several factors, including low-quality forage species that only grow in the warmer months, lack of grazing management and soil fertilization. As a result, despite current legal restrictions on deforestation, *caívas* have been replaced with exotic species reforestation or annual crops, threatening the maintenance of these forest remnants [4,11]. To reverse this situation, technologies developed by Epagri (Agricultural Research and Rural Extension Company of Santa Catarina) have promoted a significant increase in pasture production within *caívas*, with a consequent increase in animal productivity, through the adoption of techniques such as soil correction and fertilization, overseeding with annual winter forages, control of grazing, and replacement of naturalized pastures with cultivated forages, without harming forest regeneration or maintenance of the tree layer [12–14].

A fundamental step towards the adoption of these technologies is the selection of the areas within *caívas* where the level of shade provided by the trees is naturally low to moderate. To facilitate this selection, *caívas* are classified according to the density of trees (and consequent level of shade) into (i) closed *caívas*, (ii) open *caívas*, (iii) very open *caívas*, and (iv) *potreiros* [14] (see Figure 1). Epagri technologies should only be adopted in areas where shading does not exceed 60%, that is, in open or very open *caívas* or in pastures.

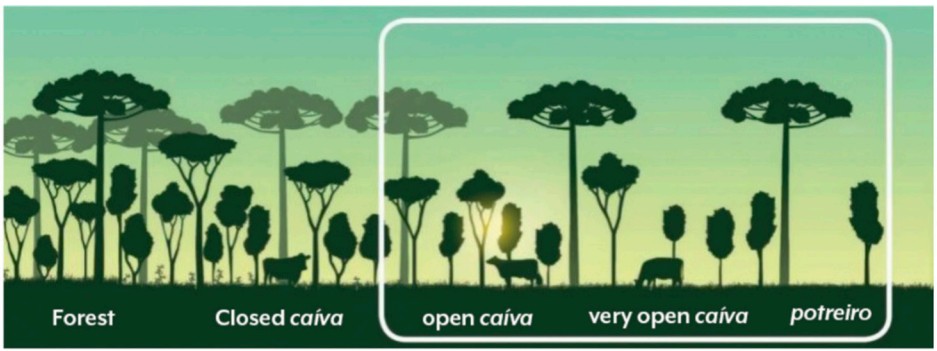

**Figure 1.** Variation in shading levels in *caívas* with different levels of forest cover, from a forest area (with many trees) to open pasture area with few trees ("*potreiro*") (source: adapted from Marques [14]).

The adoption of technologies proposed by Epagri has led to an increase in pasture production of up to 400%; with the adoption of pasture improvement technology, pasture dry mass productivity (DM) has increased from 2.5 t·ha$^{-1}$·year$^{-1}$ observed in traditional *caívas* to 10 t·ha$^{-1}$·year$^{-1}$ [2]. However, over time, this management has also affected native tree growth, with a consequent increase in the level of shading. In turn, tree growth has had an impact on the maintenance of the pasture in the system, since high levels of shade are considered the factor that most affects the productivity of forage in the system [15,16].

The current research challenge is to establish a forest management methodology that maintains the initial level of shading in the *caíva* areas that adopt pasture improvement technologies, ensuring adequate pasture production over time that is compatible with the other positive aspects of forest coverage. However, managing native trees is still a major challenge in Brazil and around the world and is one of the obstacles to the adoption of agroforestry systems in general. Forest management, with pruning and tree removal, is a sensitive topic that must be addressed through research.

In Brazil, although *caívas* are not mentioned in environmental legislation, native yerba mate production (a very similar SAF to *caívas*) and their management in forest environments are subject to two main laws: Law No. 12,651, of 25 May 2012 (New Forest Code) [17] and Law No. 11,428, of 22 December 2006 (Atlantic Forest Law) [18]. In both cases, it is possible to manage the tree layer as long as the wood is not used for commercial purposes. In legal reserves, the thinning of trees in native yerba mate systems is permitted under the Forest Code and the Atlantic Forest Law on small rural properties for use on the property and

does not require authorization. In this case, the regulatory decree establishes a limit of up to 15 m$^3$ of firewood per year and 20 m$^3$ of wood every three years [18].

As the focus of this research is the sustainability of *caívas* after the adoption of pasture improvement technologies, we sought to generate a protocol for managing native trees so that that the initial structure of the selected area and the conditions for its perpetuation were maintained. This was achieved through pruning and removal within the limits permitted by environmental legislation.

In this context, this study aims to provide a better understanding of forest dynamics in response to tree management through pruning and controlled removal in a *caíva* that has implemented technologies to increase pasture production for more than ten years. To this end, we analyze native vegetation parameters over time to validate a management proposal that does not significantly interfere in the forest dynamics of the production system and maintain native species coverage in accordance with the biome in which it is located, as required by law in Brazil. Based on the results obtained, we discuss the possibilities for conservation and sustainable use of these remnants to maintain healthy and diverse forests while also being used for productive purposes.

## 2. Materials and Methods

### 2.1. Characterization of the Study Area

This case study was carried out on a small-scale farm (22 ha) in Canoinhas, Santa Catarina State, Brazil (26°13′23″ S and 50°27′7″ W), climate Cfb according to the Köeppen classification [2], and 810 m above sea level, with a predominance of Oxisols (Figure 2). The farm has a variety of productive systems including annual crops (soy and corn), dairy cow production in managed pastures, and a 10 ha *caíva* with dairy cow grazing and native yerba mate trees, where the experiment was conducted.

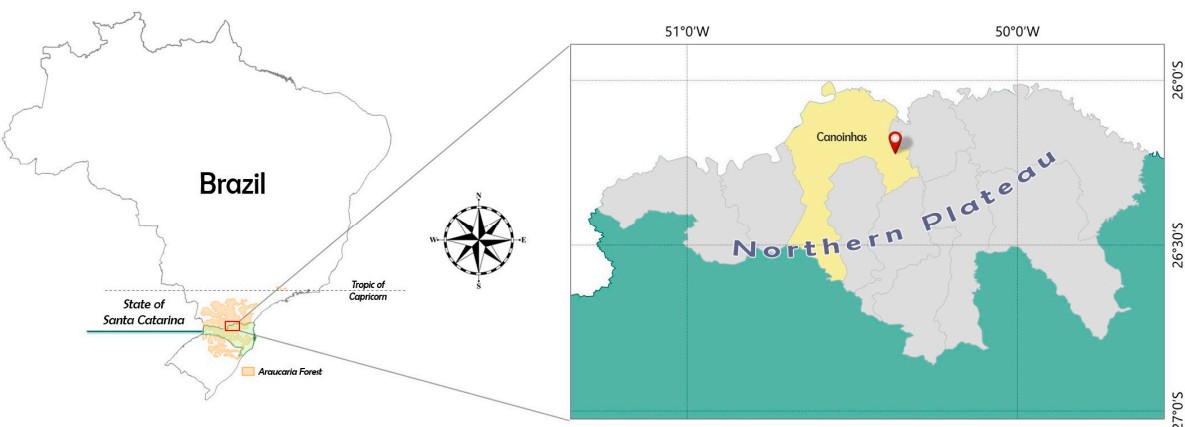

**Figure 2.** Location of experimental area within Santa Catarina state, Brazil, in the Northern Plateau region and municipality of Canoinhas.

The forest in the *caíva* has never been clear-cut, although selective logging took place approximately 30 years ago and focused on valuable timber species (*Araucaria angustifolia* and *Ocotea porosa*, among others). The understory is cleared annually by mowing to facilitate yerba mate harvesting and animal grazing. Periodically, trees are pruned or removed to provide light for pasture and yerba mate development (method defined empirically by the producers). This forest management historically carried out in the property's *caíva* contributed to the maintenance of high density tree cover and tree diversity (Figure 3).

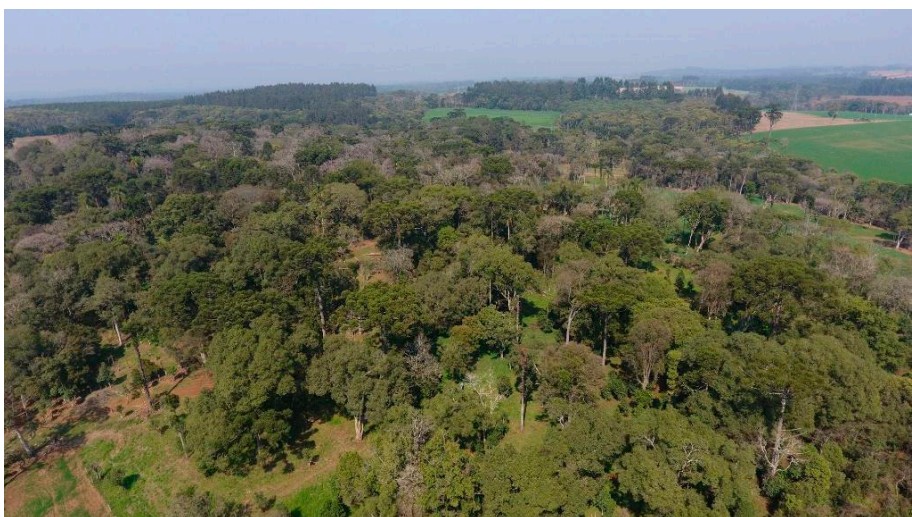

**Figure 3.** Drone photo of *caíva* on the farm where the 0.42 ha experimental area is located. Canoinhas, SC, Brazil (source: Jonatan Jumes 2023).

*2.2. Developing a Forest Management Methodology for Caívas with Improved Pasture*

To construct the tree layer management proposal for *caívas* that implements pasture improvement technologies proposed by Epagri—requiring shade levels of less than 60%—the first phase was to conduct consultations with environmental regulatory bodies, so that the main options to address the needs of the study were identified. After meetings with institutions such as the Santa Catarina Environmental Institute (IMA), Chico Mendes Institute for Biodiversity Conservation (ICM-Bio), Embrapa Forestry, and Epagri, there was a consensus in adopting the standards presented in Normative Instructions 25/2021 and 26/2021 of IMA/SC [19,20], which allows for the harvesting of up to 15 m$^3$ of firewood per year on small-scale family farms, as outlined in the Atlantic Forest Law.

Based on this premise, a forest management methodology was proposed with the following steps, which must be performed periodically by technicians or producers to maintain ideal conditions for improved pasture development:

1. Periodic assessment of shade levels and/or canopy coverage. When there is more than 60% coverage, continue to step 2;
2. Conduct a floristic survey (identification) of tree species in the area where tree management will be performed;
3. Count the number of trees of each species;
4. Start forest management by pruning the lowest branches of trees that are providing the greatest shade;
5. After pruning, if shade levels remain above 60%, begin to remove trees based on the following criteria: (a) dead trees or those in a severe state of degradation; (b) trees with a large number of individuals of the same species in the area;
6. **Exclude from management any species on the endangered species list** (*Ocotea porosa*, *Araucaria angustifolia*, *Cedrela fissilis*, *Ocotea puberula*, *Ocotea catharinensis*), with the exception of pruning lower branches (permitted by legislation);
7. **Never remove trees that only have one individual of the species;**
8. Do not exceed 15 m$^3$ of firewood/year/property;
9. After management, measure the light intensity and carry out a new survey of the species and number of individuals to ensure that management has not changed the phytosociological indicators;
10. **Do not use**, under any circumstances, removed or pruned trees **for commercial activities or sale**. According to current legislation, they can only be used for domestic use or left to decompose in stacks in the area.

It is important to highlight that there is considerable natural variation in forest cover in the region's caívas. Therefore, there is no way to define a fixed period between the assessments proposed in step 1. In general, farmers notice when the level of shading increases over time and should start monitoring from this moment on. Furthermore, the species cited in step 6 are considered vulnerable on the list of endangered species of the Brazilian Institute of the Environment and Renewable Natural Resources (IBAMA) and more recently, also as critically endangered (CR) on the Red List of Threatened Species of the International Union for Conservation of Nature (IUCN) [21].

For this study, this methodology was applied to an area of 0.42 ha that began implementing pasture improvement technology in 2013. The evaluation began in February 2020, when a measurement of shade levels was conducted. We found that shade levels were above 80% and already affecting pasture growth and yerba mate present in the understory (Figure 4a). In March 2020, management was carried out according to the methodology proposed in this study, with pruning and removal of some individuals until obtaining a shade level of 60% (Figure 4b,c). Then, the pasture improvement management proposed by Epagri was continued (Figure 4d).

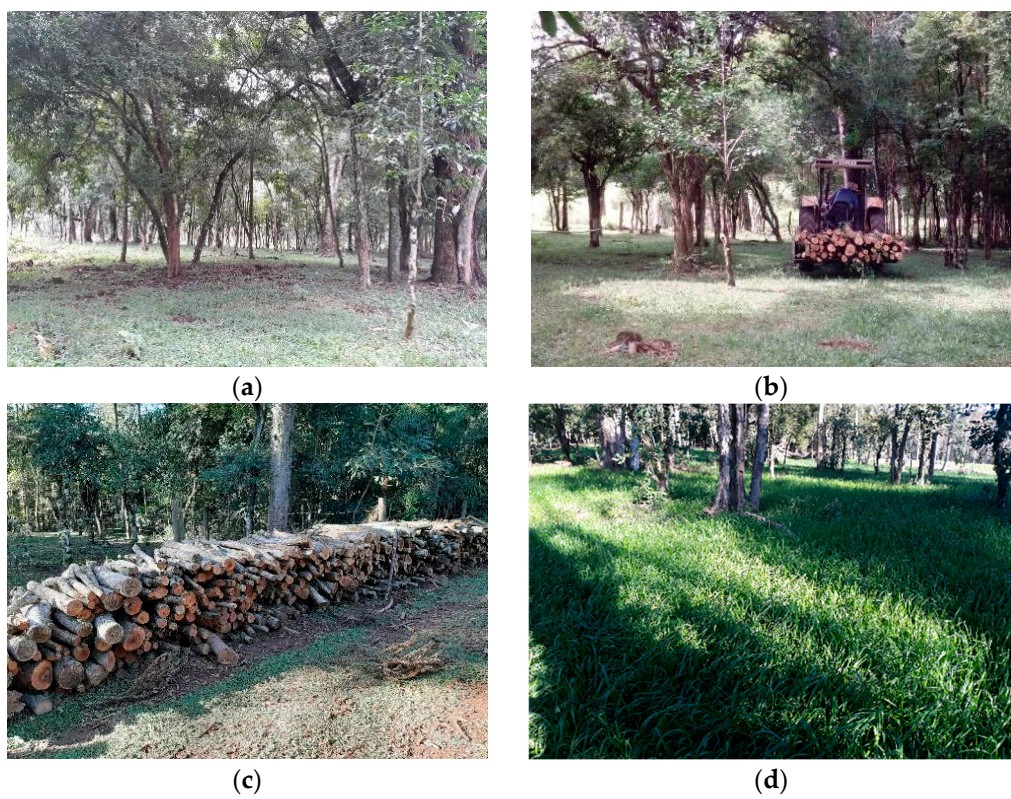

**Figure 4.** *Caíva* area: (**a**) in 2020 with excess shading, affecting the pasture implemented in 2013; (**b**) tree management with pruning and removal of some trees in accordance with the methodology proposed herein, conducted in 2020; (**c**) 13 m$^3$ of firewood stacked for use on the farm; (**d**) well-developed winter pasture three months after tree management. Canoinhas, SC, Brazil.

Two years after the first intervention in the *caíva*, during which 13 m$^3$ of firewood was removed, a new management process was conducted with the same methodology to remove another 8 m$^3$ and maintain ideal light levels. The indicators used to evaluate the efficiency of the methodology to reduce shade levels and maintain forest cover and diversity indices within the studied *caíva* are described below.

*2.3. Data Collection and Analysis*

2.3.1. Forest Indicators

Floristic and phytosociological studies provide important information for understanding local and regional diversity by identifying the biogeographic patterns of plant communities [22]. Thus, to better understand the structure of the forest community, a floristic and phytosociological study of the adult arboreal component of the *caíva* was conducted.

Assessments were carried out in September 2013 (initial adoption of pasture improvement technologies), February 2020 (before the first forest management), August 2021 (to monitor the arboreal stratum), and May 2022 (after the second forest management). The experimental unit with a total area of 4320 m$^2$ was subdivided into 36 contiguous plots of 10 m × 12 m. In each plot, all arboreal and palm tree individuals with a diameter at breast height (DBH) ≥ 5.0 cm and height > 1.30 m were identified, measured, and mapped using a coordinate system.

In order to evaluate the level of influence of human activity on the structure and floristic composition of the remnant, the data were used to quantify the following phytosociological parameters: frequency, density, dominance, importance value, and basal area (*sensu* Mueller-Dombois & Ellenberg [23], units indicated in results), calculated using the Fitopac 2.1 software [24].

For species identification, scientific names were based on the APG IV system [25]; confirmation of the species found and an update of their botanical nomenclatures were carried out using the Tropicos [26] and Flora e Funga do Brasil [27] websites. Information on successional status and those related to the species dispersion were obtained in Meyer et al. [28]. Following Budowski [29], we classified species into ecological groups as pioneer, secondary, and late successional.

To evaluate vertical stratification, the categories presented in the IBGE [30] were used, defined by classes of height, namely: macro-phanerophyte (≥30 m), meso-phanerophyte (≥20 m <30 m), micro-phanerophyte (≥5 m <20 m), and nano-phanerophyte (>0.25 m < 5 m). For this purpose, the maximum expected height for the species when adult (potential height) was obtained from the Flora e Funga do Brasil website [27], Meyer et al. [28], Pinotti, Hanisch & Negrelle [5], and Carvalho [31–35], as well as data on the phytogeographical domains of the species and their successional status.

The evaluation of the floristic similarity among the areas was performed using the binary Sørensen Index (IS$_\varnothing$), the diversity was estimated by Shannon Index (H′), and the distribution of the abundance by the Pielou evenness index (J′) [28]. To verify whether there was a difference among the diversities obtained in each survey, the H′ values were compared pairwise using the Hutcheson *t*-test, with significance *p*-value < 0.05 (5%) and $t_{tab}$ = 1.96 [36].

2.3.2. Measuring Shade Level from PAR Data

Photosynthetically active radiation (PAR) was measured between 11:00 h and 13:00 h, with clear skies, using a portable digital PAR meter and *LightScout*® photosynthetic photon flux (PPF; μmol·m$^{-2}$·s$^{-1}$) at 1 m above ground level. The measurements began in February 2020, before the management of the tree strata of the *caíva*, and occurred at least once per season until August 2022.

2.3.3. Measuring Soil Attributes

As the *caíva* was fertilized annually from May 2020, the chemical attributes of the soil were monitored over the three years of evaluation. Soil samples (0–10 cm deep) were taken for nutrient and pH assessment [37]. Dolomitic lime (4 t·ha$^{-1}$·year) was applied directly onto the soil to achieve a target pH of 5.5; fertilizer was applied twice a year (Abril and October) (80 kg P$_2$O$_5$ + 60 kg K$_2$O + 100 kg·ha$^{-1}$·year) to adjust soil nutrients to an average fertility level based on the recommendations for Warm Season Grasses [38].

### 3. Results and Discussion

Here, we show the results of a relatively simple methodology for managing the arboreal component in SAFs with native trees, to maintain shade levels at around 50–60%, distributed over the total area. To achieve this, the phytosociological indicators of the area were compared before and after adopting the methodology.

### 3.1. Forest Indicators

The number of tree individuals in 2013, when the adoption of pasture improvement technology began, was 636 trees·ha$^{-1}$, with a diversity of 18 species typical of the Araucaria Forest and a basal area of 11 m$^2$·ha$^{-1}$. The *caíva* was classified as an "open *caíva*" [14]. Even with the use of annual fertilization, overseeding of pastures, and controlled grazing, over eight years we found no negative effects on forest indicators. On the contrary, an increase was observed in the number of individuals (673 trees·ha$^{-1}$), number of species (25), and basal area, which explains the increase in shade in the study area (Tables 1, A1 and A2).

**Table 1.** Floristic characteristics documented in each survey. Canoinhas, SC, Brazil.

| Year | AD (ni·ha$^{-1}$) | S | H′ | Var (H′) | J′ | BA (m$^2$·ha$^{-1}$) |
|------|------|----|------|--------|------|------|
| 2013 | 636.57 | 18 | 1.37 | 0.0079 | 0.47 | 11.38 |
| 2020 | 673.61 | 25 | 1.99 | 0.0059 | 0.62 | 12.22 |
| 2021 | 495.37 | 26 | 2.13 | 0.0090 | 0.65 | 10.09 |
| 2022 | 481.48 | 26 | 2.17 | 0.0086 | 0.67 | 10.65 |

AD = absolute density; ni = number of individuals; S = specific richness (number of species); H′ = Shannon's diversity index; var = variance; J′ = Pielou's evenness; BA = basal area.

The increases in species richness and diversity indices between 2013 and 2020 were very positive results that validate the use of Epagri's pasture improvement technologies in *caívas* as strategies to increase animal productivity and support environmental conservation. The results indicate that even though they were used intensively throughout the entire period, pasture management and cattle grazing did not negatively affect the arboreal strata. The increase in diversity indices indicates the occurrence of forest regeneration and growth of existing trees. These data corroborate the results observed in [13] that cattle, in controlled grazing systems, do not consume the tree regeneration in *caívas*.

On the other hand, these results also prove that despite the positive aspects for environmental conservation, the increase in the floristic characteristics of the *caíva* tree strata can modify the characteristics of the area as an open or very open *caíva* silvopastoral system. In this case, high tree density can harm pasture development, which justifies intervention with tree management.

The results for floristic indicators after applying the forest management methodology proposed herein confirmed the efficacy of the method. We found a reduction in the number of individuals, but species richness and diversity indices and basal area remained consistent, both after the first intervention in 2020 when 13 m$^3$ of firewood was removed and after the second management of the tree layer in 2022 (Table 1).

Both the Hutcheson *t*-test comparison matrix and the similarity matrix indicated that there was no significant effect on the similarity indexes over the years (Tables 2 and 3). The similarity between the years showed high values between them, above 80%, indicating that despite management involving the removal of some individuals, it did not affect forest conservation of the surrounding area over the years (Table 3).

Over the years of evaluation, the occurrence of 26 species distributed across 15 botanical families that are characteristic of the Araucaria Forest was maintained, including two species that are registered on the list of endangered species [28] (Table 4).

**Table 2.** Hutcheson's *t*-test matrix results ($t_{calc}$) for H′ diversity among surveys. Canoinhas, SC, Brazil.

| Year | 2013 | 2020 | 2021 | 2022 |
|------|------|------|------|------|
| 2013 | -- | | | |
| 2020 | −5.33 | -- | | |
| 2021 | −5.81 | −1.07 | -- | |
| 2022 | −6.23 | −1.44 | −0.33 | -- |

Degrees of Freedom = ∞; *p*-value < 0.05 (5%); $t_{tab}$ = 1.96.

**Table 3.** Similarity measure of $IS_\varnothing$ similarity matrix among surveys. Canoinhas, SC, Brazil.

| Year | 2013 | 2020 | 2021 | 2022 |
|------|------|------|------|------|
| 2013 | -- | | | |
| 2020 | 0.84 | -- | | |
| 2021 | 0.82 | 0.98 | -- | |
| 2022 | 0.82 | 0.98 | 1.00 | -- |

The results presented herein are similar to surveys from other *caívas* in the Santa Catarina Northern Plateau, where the number of species varied between 18 and 42 [7]. According to Lacerda [39], forests under traditional management with yerba mate have high levels of diversity and, when assessed at the landscape scale, preserve a significant amount of the tree species diversity that is currently found in the Araucaria Forest. This value represents between 20 and 30% of the number of species in the pilot inventory of the Santa Catarina Forest and Floristic Inventory, which was 133 species [40]. *Caívas*, as SSPs with native trees, in addition to effectively contributing to the conservation of tree species, play an important role in the region as connectivity corridors between remnants [14]. Recent studies have indicated that community-managed forests outside protected areas can provide not only better forest coverage but also other conservation benefits [41].

Although distribution of abundance was not a criteria adopted in the forest management proposal, it remained similar across all surveys, representing the form of an inverted J which is typical of managed Araucaria Forests (Figure 5 and Table 4). We can see that the increase in the number of tree species over the years occurred due to the entry of new individuals. This indicates that, even with animal grazing and the adoption of technologies such as fertilization, overseeding of forage species and grazing control, the forest regeneration continues to be active in the *caívas*, confirming the results obtained in previous research [13].

**Table 4.** Floristic composition, density, ecological group, and life form, recorded for the arboreal component in each survey. Canoinhas, SC, Brazil.

| Family *Species* | Regional Common Name | Number of Individuals 2013 | 2020 | 2021 | 2022 | EG | LF |
|------------------|----------------------|------|------|------|------|----|----|
| **Annonaceae** | | | | | | | |
| *Annona neosalicifolia* H.Rainer | Araticum amarelo | 4 | 8 | 7 | 6 | P | Meso |
| *Annona rugulosa* (Schltdl.) H.Rainer | Araticum preto | 8 | 17 | 15 | 16 | P | Micro |
| **Aquifoliaceae** | | | | | | | |
| *Ilex brevicuspis* Reissek | Voadeira | 5 | 5 | 5 | 7 | S | Meso |
| *Ilex paraguariensis* A.St.-Hil. | Erva-mate | 180 | 128 | 94 | 83 | P | Macro |
| *Ilex theezans* Mart. ex Reissek | Caúna; Congonha | - | - | 1 | 1 | S | Meso |
| **Araucariaceae** | | | | | | | |
| *Araucaria angustifolia* (Bertol.) Kuntze | Araucaria | 3 | 3 | 3 | 3 | P | Macro |
| **Arecaceae** | | | | | | | |
| *Syagrus romanzoffiana* (Cham.) Glassman | Palmeira Jerivá; Jerivá | 1 | 1 | 1 | 1 | P | Macro |
| **Canellaceae** | | | | | | | |
| *Cinnamodendron dinisii* Schwacke | Pimenteira | 1 | 2 | 2 | 2 | P | Meso |
| **Combretaceae** | | | | | | | |
| *Terminalia australis* Cambess. | Sarandi | - | 1 | 1 | 1 | P | Micro |

**Table 4.** *Cont.*

| Family *Species* | Regional Common Name | Number of Individuals | | | | EG | LF |
|---|---|---|---|---|---|---|---|
| | | 2013 | 2020 | 2021 | 2022 | | |
| **Euphorbiaceae** | | | | | | | |
| *Gymnanthes klotzschiana* Müll.Arg. | Branquilho | - | 1 | 1 | 1 | P | Micro |
| *Sapium glandulosum* (L.) Morong | Leiteiro | 8 | 10 | 6 | 6 | P | Meso |
| **Fabaceae-Faboideae** | | | | | | | |
| *Lonchocarpus nitidus* (Vogel) Benth. | Timbó; Timbózinho | - | 2 | 2 | 2 | S | Micro |
| *Machaerium* Pers. | Pau-ferro; Pau-marfim | 1 | 1 | 1 | 1 | S | Meso |
| **Lauraceae** | | | | | | | |
| *Nectandra megapotamica* (Spreng.) Mez | Canela fedorenta | 2 | 2 | 1 | 1 | S | Meso |
| *Ocotea porosa* (Nees & Mart.) Barroso | Imbuia | 12 | 12 | 14 | 14 | P | Macro |
| *Ocotea puberula* (Rich.) Nees | Canela guaicá | 3 | 3 | 1 | 1 | P | Meso |
| **Meliaceae** | | | | | | | |
| *Cedrela fissilis* Vell. | Cedro; Cedro rosa | 1 | 1 | 1 | 1 | S | Macro |
| **Myrtaceae** | | | | | | | |
| *Campomanesia xanthocarpa* (Mart.) O.Berg | Guabiroba | - | 2 | 1 | 1 | S | Meso |
| *Curitiba prismatica* (D.Legrand) Salywon & Landrum | Cerninho | 39 | 60 | 29 | 36 | S | Micro |
| *Eugenia uniflora* L. | Pitanga | - | 1 | 1 | 1 | P | Micro |
| *Myrceugenia myrcioides* (Cambess.) O.Berg | Guamirim | 3 | 11 | 14 | 12 | C | Micro |
| | | | | | | | Continue... |
| **Rutaceae** | | | | | | | |
| *Zanthoxylum rhoifolium* Lam. | Mamica de cadela | - | 1 | 1 | 1 | S | Micro |
| **Salicaceae** | | | | | | | |
| *Casearia decandra* Jacq. | Guaçatunga; Guaçatunga branca | 2 | 10 | 5 | 4 | S | Micro |
| *Casearia sylvestris* Sw. | Guaçatunga preta | - | 2 | 2 | 2 | S | Meso |
| **Sapindaceae** | | | | | | | |
| *Allophylus edulis* (A.St.-Hil. et al.) Hieron. ex Niederl. | Vacum | 1 | 3 | 4 | 3 | S | Micro |
| **Winteraceae** | | | | | | | |
| *Drimys brasiliensis* Miers | Cataia | 1 | 1 | 1 | 1 | S | Meso |
| Total individuals per year | | 275 | 291 | 214 | 208 | | |

EG = ecological group (P = pioneer, S = secondary, C = climax); LF = life form (Macro = macro-phanerophyte, Meso = meso-phanerophyte, Micro = micro-phanerophyte).

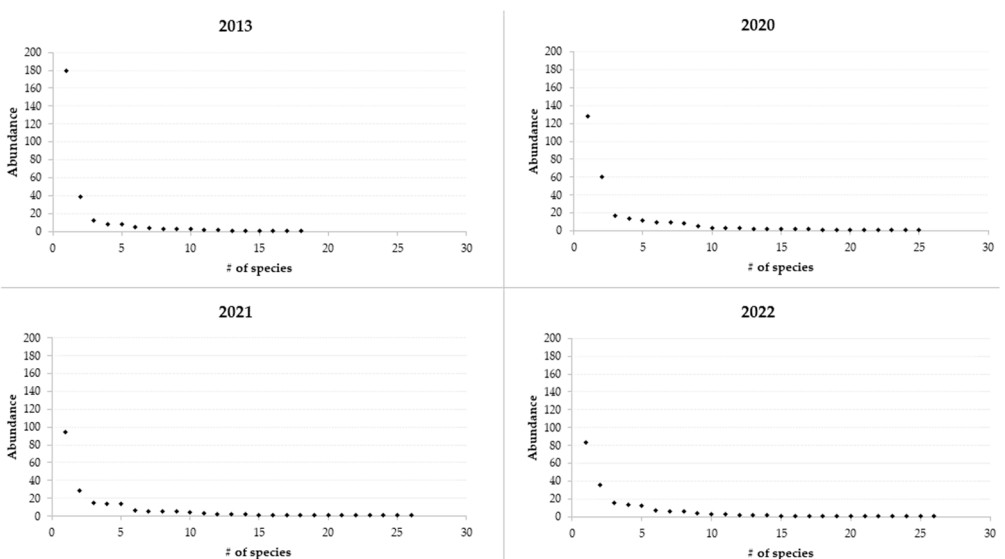

**Figure 5.** Individual abundance distribution of the different species identified among surveys of the arboreal stratum of the *caíva* after interventions. Canoinhas, SC, Brazil.

Ecological groups were maintained, with a predominance of pioneers and secondary species (Figure 6 and Table 4), as were life forms, with a predominance of meso and microphanerophyte species (Figure 7 and Table 4).

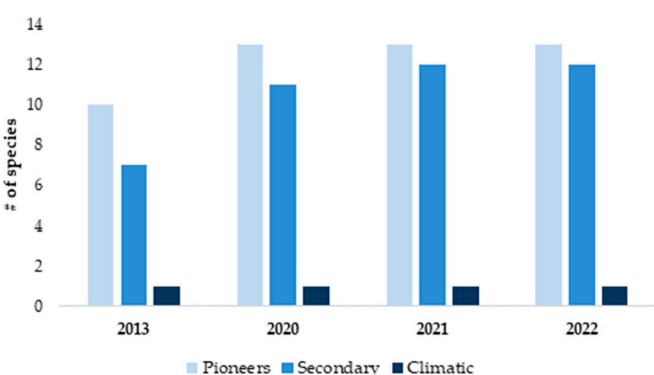

**Figure 6.** Variation in the number of species per ecological group over the survey years in the *caíva* with improved pasture and arboreal stratum management. Canoinhas, SC, Brazil.

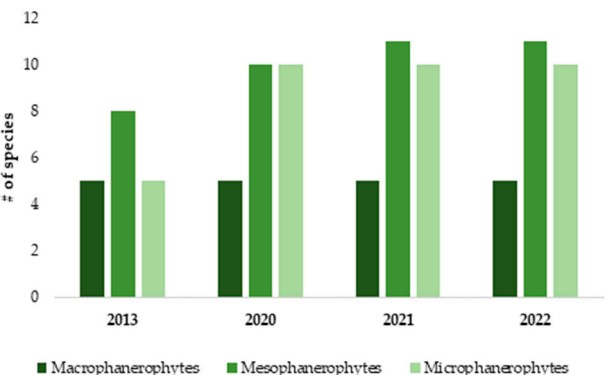

**Figure 7.** Variation in the number of species per life form over the survey years in the *caíva* with improved pasture and arboreal stratum management. Canoinhas, SC, Brazil.

As with the other evaluated indicators, even after pruning and controlled removal of trees in the *caíva* following the tree management methodology, there was no effect on the structural characteristics of the tree strata. In other words, the *caíva* selected in 2013 to introduce pasture improvement technologies maintained the same structural characteristics after tree management, with trees with an average diameter of 18 cm and an average height between 6 and 7 m and some larger individual trees (Table 5).

**Table 5.** Structural characteristics of the tree strata of a *caíva* with the adoption of technologies for pasture improvement in 2013 (soil fertilization, grazing control, introduction of improved forages), eight years later (2020), and after applying two management regimes to the tree strata (2021 and 2022). Canoinhas, SC, Brasil.

| Year | Diameter (cm) | | | | Height (m) | | | |
|------|---------------|--|--|--|------------|--|--|--|
| | $\mu \pm \sigma$ | | Maximum | Minimum | $\mu \pm \sigma$ | | Maximum | Minimum |
| 2013 | 18.2 | 14.0 | 96.1 | 3.8 | 5.9 | 5.6 | 28.0 | 1.5 |
| 2020 | 17.7 | 14.9 | 98.7 | 1.8 | - | - | - | - |
| 2021 | 18.0 | 16.6 | 112.8 | 4.1 | - | - | - | - |
| 2022 | 19.0 | 17.1 | 98.9 | 4.9 | 7.3 | 5.5 | 29.5 | 2.0 |

$\mu$ = mean; $\sigma$ = standard deviation.

The most important change in the tree strata among the 2013 and 2022 surveys was the importance value (IV) of yerba mate. The IV went from 124 in 2013, with 417 individuals, to 77 in 2022 and 192 individuals, although yerba mate maintained its position as the species with the highest IV in the area (Tables A1 and A2). The importance value expresses the ecological relevance of the species in a community, in terms of horizontal distribution,

and its estimation is based on the relationship between the relative indices of density, dominance, and frequency of the sampled species [24].

This sharp reduction in yerba mate, however, occurred partly as a management strategy. In 2013, there was a large number of diseased or dead yerba mate trees, most of which occurred in "*reboleiras*" (a cluster of trees in a very close area); thus, it was proposed to maintain the most vigorous trees and eliminate those in an advanced state of degradation.

For the other species, the phytosociological characteristics remained very similar over the analyzed years, indicating that the methodology proposed for this study was effective in maintaining the arboreal structure of the *caíva* consistent with the initial conditions when the pasture improvement technology was implemented.

### 3.2. Shade Levels from PAR Data

Figure 3 shows the heterogeneity of the tree canopy in the *caíva*, which provides a good representation of the *caívas* in the Northern Plateau region. This variation in the crowns of different species and trees with different heights directly affects the measurement of light intensity in the lower stratum of this traditional silvopastoral system, promoting varying levels of shade. This, in turn, is reflected in the productivity of the plants present in the system.

In the first assessment of shade levels in February 2020, the canopy coverage was so "closed" that there was practically no variation across sampling points. The average PAR reaching the herbaceous stratum was 100 $\mu$mol$\cdot$m$^{-2}\cdot$s$^{-1}$ (Figure 8), a value which makes the development of pasture species proposed in *caíva* improvement technology unfeasible. By way of comparison, the PAR in full sun at the same time was 1650 $\mu$mol$\cdot$m$^{-2}\cdot$s$^{-1}$. This measurement was fundamental for decision making for tree management, and the effect can be seen in the following measurements where the results indicated a greater variation in points of high light intensity in the area.

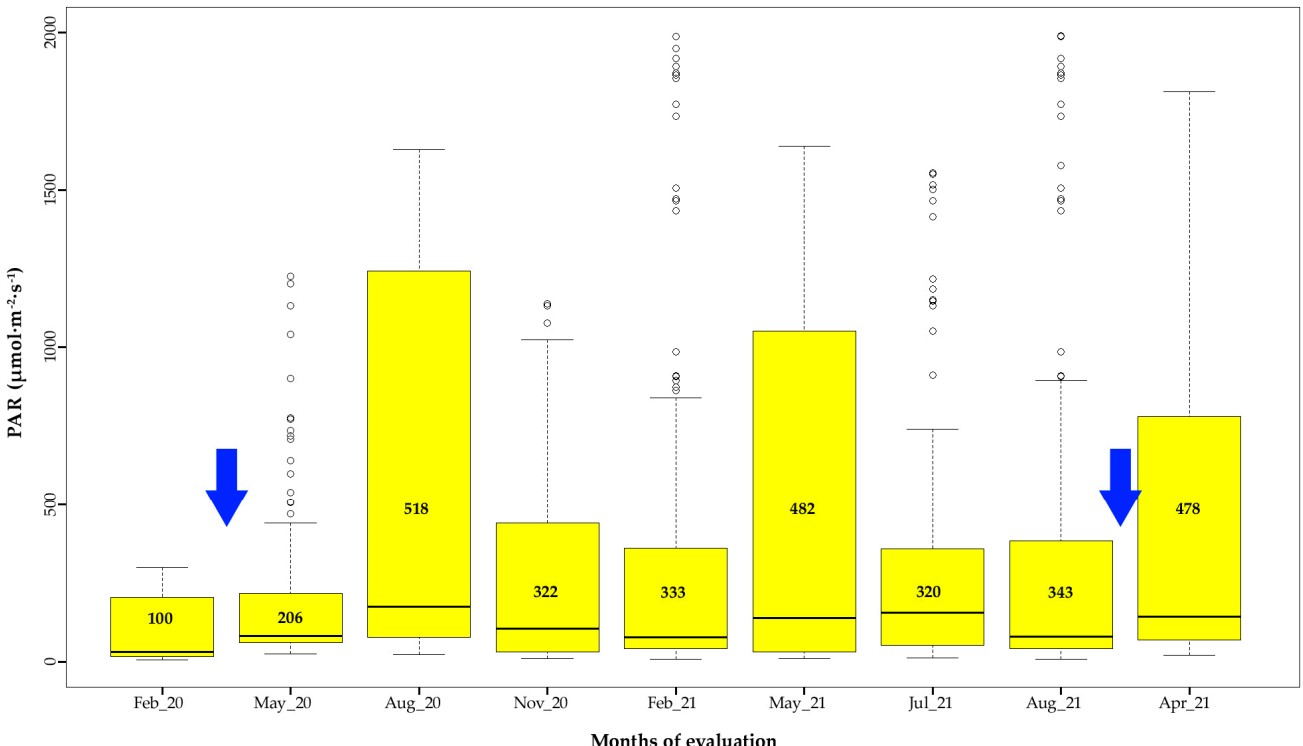

**Figure 8.** Box-plot distribution and average of the photosynthetically active radiation (PAR) over 26 months of evaluation in the *caíva* with pasture improvement. Tree layer management occurred in March 2020 and March 2022 (blue arrows). Box-plots and whiskers represent quartiles, median, upper, and lower limits; ○ = outliers. Canoinhas, SC, Brazil.

After carrying out the first tree management (March–April 2020), with pruning and tree removal, although the average remained low, it was possible to observe many points (outliers) of intense radiation in the area, which increased over time, Thus, the expected results were achieved with management. However, just over a year later, we found the need to apply another management regime in order to prevent the shade levels from returning to the 2020 levels. Therefore, tree management was again carried out in March 2022, which quickly promoted the necessary increases in light to an average of 478 $\mu$mol$\cdot$m$^{-2}\cdot$s$^{-1}$ in April 2022 (Figure 8), suitable for the development of pastures present in the region.

The PAR values obtained throughout the year in this area after the interventions are similar to the values observed by other authors in areas that adopt *caíva* improvement technology. On average, PAR values are 70% throughout the year, even in those considered open and more open *caívas* in the region, ranging from 50 to 75% [10,13].

### 3.3. Soil Indicators

Monitoring soil fertility is essential to adjust fertilization in pastures established in improved fields. In this study, the results confirmed the efficacy of the method proposed by Epagri of dividing the use of inputs and application as cover, without incorporation into the soil, to increase pH and other fertility indicators (Table 6).

**Table 6.** Soil attributes of the experimental area over the years of evaluation of the *caíva* with pasture improvement technology and tree stratum management. Canoinhas, SC, Brasil.

| Attributes | 2020 | 2021 | 2022 |
|:---:|:---:|:---:|:---:|
| Clay (%) | 39 | 34 | 40 |
| pH$_{water}$ | 4.4 | 4.9 | 5.1 |
| P (mg$\cdot$dm$^{-3}$) | 3.9 | 3.9 | 4.3 |
| K (mg$\cdot$dm$^{-3}$) | 119 | 113 | 118 |
| SOM (%) | 1.9 | 2.2 | 2.3 |
| Al (cmol$_c\cdot$dm$^{-3}$) | 3.8 | 1.4 | 0.6 |
| Ca (cmol$_c\cdot$dm$^{-3}$) | 2.1 | 4.9 | 6.3 |
| Mg (cmol$_c\cdot$dm$^{-3}$) | 0.7 | 1.8 | 2.3 |
| V% | 21 | 65 | 76 |

P = phosphorus; K = potassium; SOM = soil matter organic; Al = aluminum; Ca = calcium; Mg = magnesium; V% = soil base saturation.

An interesting observation in Table 6 is that although the study area has been using fertilizers since 2013, in 2019, the owner stopped applying them since even with fertilization, the pastures were not developing due to excess shade.

Not applying fertilizers had a direct effect on soil pH, which returned to the 2013 levels (pH = 4.4) in the 0–10 cm surface layer. However, as soon as the limestone was applied again, the pH value increased. In other words, to maintain silvopastoral systems or other productive agroforestry systems, the proposed technologies must be used correctly and for the entire period in which the system is being used intensively. The application of inputs to improved *caívas* is designed to meet the nutritional demand of the pastures that form the herbaceous stratum. It is performed so gradually, on the surface and in small quantities, so as to not cause drastic changes to the system.

The challenge of finding a balance between economic and ecological demands in the use of landscapes is an important consideration for achieving productive systems that conserve natural resources and generate income for the families that maintain them. However, it would be naive to think that the adoption of technologies that increase the productivity of traditional SAFs, such as *caívas*, can occur without management and/or interference in the tree strata.

For more than a century, the use of traditional management systems has proven that it is possible to maintain forest cover through use, and this use almost always includes forest management. Currently, owners of traditional SAFs face competing challenges related to the need to improve income generation from these systems while ensuring forest

management remains within legal restrictions. In other words, instead of being rewarded for conserving the forest landscape, those who had have done so are left with unproductive areas on the property that offer poor income.

We would like to emphasize that productivity in *caívas* that adopt pasture improvement technology has ranged from 10 to 12 t·ha$^{-1}$ of dry mass per year [12,13], which promoted an average milk production of 8 to 10 t·ha$^{-1}$·year$^{-1}$ in forest remnants with conservation of the tree strata. This is a very positive result for both environmental conservation and income generation for families who use *caívas*. The central issue of this study, therefore, was to develop management practices that enable this system to continue functioning, without reducing the production of pasture due to excessive shade. Whenever the tree canopy grows and provides more shade, pasture production affects the property's economic income, putting the continuation of the system at risk. A new phase of research, in which this study is integrated, will develop and adjust management protocols for the tree strata. Much research is still needed to monitor the long-term effects; however, after almost two decades of work in *caívas*, we are certain that the results will continue to be promising.

Research that values and develops technologies for traditional silvopastoral systems in remnants of native forests is still rare. Studies like this seek to offer concrete information and data about these systems and how to improve them. The lack of information generated from medium-and long-term analyses has led to a situation in which pastoral animals have been removed from forested areas in many locations around the world, even in centuries-old traditional systems. In many cases, the consequences have been negative, such as an increase in forest fires, due to the accumulation of dry mass in the herbaceous stratum of forests, as well as significant socio-environmental impacts [42,43]. Changing how we see traditional systems and enabling the integration of modern and sustainable productive strategies is key to ensure that they continue to contribute to nature conservation.

## 4. Conclusions

Forest management aimed at maintaining initial shade levels of 50 to 60% in *caívas* that adopt pasture improvement technologies was effective in achieving the expected results.

No reduction was observed in species diversity and richness and a high similarity in the tree component was maintained across the 2013 and 2022 surveys, even with the adoption of the proposed forest management strategies.

The levels of management permitted in the environmental legislation (i.e., removing up to 15 m$^3$ of wood per property every two years, for non-commercial use), if performed following the criteria proposed herein, contribute to maintaining productive pasture within *caívas* and do not harm the maintenance of the forest strata.

Publicly funded research has sought to develop technologies that improve the situation of caívas, and although such long- and medium-term studies are slow, labor-intensive, and with high-costs, they are necessary.

In southern Brazil, the forest cover that remains has been conserved in part because of the application of environmental legislation in recent decades; however, farming families who have conserved the remaining areas through their use are also responsible for the continuation of the forest landscape. Adopting technologies that promote increased pasture production in *caívas* is already proving to be effective in increasing income and promoting controlled forest regeneration.

The adoption of forest management is yet another step that will contribute to the objective of maintaining productive *caívas*. We believe that results like those presented herein help to change the preconceptions regarding forest management; that is, forest conservation is not incompatible with more intensive modes of production, as long as it is based on techniques that promote the maintenance of the forest. The important thing is that the farmers who conserve the ecosystem are rewarded, whether through payments for environmental services or through support from research and rural extension to develop strategies that maintain forest cover without the need for deforestation, while also increasing the productive and economic capacities of these systems.

**Author Contributions:** Conceptualization and methodology, A.L.H.; formal analysis, L.C.A.P.; investigation, A.L.H.; writing—original draft preparation, A.L.H.; writing—review and editing: A.L.H. and L.C.A.P.; project administration, A.L.H.; funding acquisition, A.L.H. All authors have read and agreed to the published version of the manuscript.

**Funding:** This research was made possible through the financial and technical support of Epagri—Empresa de Pesquisa Agropecuária e Extensão Rural de Santa Catarina, regarding the Project nº 6314698 entitled Management of the tree layer in *caíva* areas to control shading, carried out from 10/2019 to 12/2022.

**Data Availability Statement:** Data can be made available on request.

**Acknowledgments:** We thank Miguel and Raquel Gurzynski for providing the area for this study and for their always helpful insights throughout the research period. We would also like to thank André Lacerda (Embrapa Forestry) for the partnership and Anésio da Cunha Marques (ICMBio) and the technicians from IMA Canoinhas for their support on environmental legislation in the discussion of the proposal.

**Conflicts of Interest:** Ana Lúcia Hanisch is employed by the company EPAGRI—Canoinhas Experimental Station. The authors declare that this study received funding from Epagri—Empresa de Pesquisa Agropecuária e Extensão Rural de Santa Catarina. The remaining author declares that the research was conducted in the absence of any commercial or financial relationships that could be construed as a potential conflict of interest.

## Appendix A

**Table A1.** Phytosociological characteristics of the species sampled in the initial survey (2013) ranked by importance value (IV). Canoinhas, SC, Brazil.

| Species | IV | AD (ind·ha$^{-1}$) | RD (%) | AF (%) | RF (%) | ADo (m$^2$·ha$^{-1}$) | RDo (%) |
|---|---|---|---|---|---|---|---|
| *Ilex paraguariensis* | 124.00 | 416.67 | 65.45 | 100.00 | 32.14 | 6.96 | 26.41 |
| *Ocotea porosa* | 45.59 | 27.78 | 4.36 | 27.78 | 8.93 | 8.51 | 32.30 |
| *Curitiba prismatica* | 45.43 | 90.28 | 14.18 | 66.67 | 21.43 | 2.59 | 9.82 |
| *Ilex brevicuspis* | 13.22 | 11.57 | 1.82 | 13.89 | 4.46 | 1.83 | 6.94 |
| *Sapium glandulosum* | 12.99 | 18.52 | 2.91 | 22.22 | 7.14 | 0.77 | 2.94 |
| *Araucaria angustifolia* | 11.73 | 6.94 | 1.09 | 8.33 | 2.68 | 2.10 | 7.96 |
| *Annona rugulosa* | 10.40 | 18.52 | 2.91 | 19.44 | 6.25 | 0.33 | 1.24 |
| *Ocotea puberula* | 7.83 | 6.94 | 1.09 | 5.56 | 1.79 | 1.31 | 4.96 |
| *Annona neosalicifolia* | 5.61 | 9.26 | 1.45 | 11.11 | 3.57 | 0.15 | 0.59 |
| *Myrceugenia myrcioides* | 4.30 | 6.94 | 1.09 | 8.33 | 2.68 | 0.14 | 0.53 |
| *Cedrela fissilis* | 4.17 | 2.31 | 0.36 | 2.78 | 0.89 | 0.77 | 2.91 |
| *Nectandra megapotamica* | 3.74 | 4.63 | 0.73 | 5.56 | 1.79 | 0.32 | 1.23 |
| *Casearia decandra* | 2.74 | 4.63 | 0.73 | 5.56 | 1.79 | 0.06 | 0.23 |
| *Drimys brasiliensis* | 1.89 | 2.31 | 0.36 | 2.78 | 0.89 | 0.17 | 0.64 |
| *Syagrus romanzoffiana* | 1.87 | 2.31 | 0.36 | 2.78 | 0.89 | 0.16 | 0.62 |
| *Allophylus edulis* | 1.60 | 2.31 | 0.36 | 2.78 | 0.89 | 0.09 | 0.34 |
| *Cinnamodendron dinisii* | 1.46 | 2.31 | 0.36 | 2.78 | 0.89 | 0.05 | 0.20 |
| *Machaerium* sp. | 1.40 | 2.31 | 0.36 | 2.78 | 0.89 | 0.04 | 0.15 |

AD = absolute density; ADo = absolute dominance; RDo = relative dominance (%); RD = relative density (%); AF = absolute frequency (%); RF = relative frequency (%).

**Table A2.** Phytosociological characteristics of the species sampled in the final survey (2022) ranked by importance value (IV). Canoinhas, SC, Brazil.

| Species | IV | AD (ind·ha$^{-1}$) | RD (%) | AF (%) | RF (%) | ADo (m$^2$·ha$^{-1}$) | RDo (%) |
|---|---|---|---|---|---|---|---|
| *Ilex paraguariensis* | 77.23 | 192.13 | 39.90 | 91.67 | 23.74 | 3.35 | 13.58 |
| *Ocotea porosa* | 54.92 | 32.41 | 6.73 | 33.33 | 8.63 | 9.75 | 39.56 |
| *Curitiba prismatica* | 44.38 | 83.33 | 17.31 | 75.00 | 19.42 | 1.89 | 7.65 |
| *Annona rugulosa* | 17.80 | 37.04 | 7.69 | 30.56 | 7.91 | 0.54 | 2.19 |
| *Ilex brevicuspis* | 17.31 | 16.20 | 3.37 | 19.44 | 5.04 | 2.20 | 8.91 |
| *Araucaria angustifolia* | 15.24 | 6.94 | 1.44 | 8.33 | 2.16 | 2.87 | 11.64 |
| *Myrceugenia myrcioides* | 14.05 | 27.78 | 5.77 | 27.78 | 7.19 | 0.27 | 1.08 |
| *Sapium glandulosum* | 10.65 | 13.89 | 2.88 | 16.67 | 4.32 | 0.85 | 3.45 |
| *Annona neosalicifolia* | 8.14 | 13.89 | 2.88 | 16.67 | 4.32 | 0.23 | 0.94 |
| *Casearia decandra* | 5.39 | 9.26 | 1.92 | 11.11 | 2.88 | 0.14 | 0.59 |
| *Cedrela fissilis* | 4.81 | 2.31 | 0.48 | 2.78 | 0.72 | 0.89 | 3.61 |
| *Nectandra megapotamica* | 4.51 | 2.31 | 0.48 | 2.78 | 0.72 | 0.82 | 3.31 |
| *Allophylus edulis* | 4.25 | 6.94 | 1.44 | 8.33 | 2.16 | 0.16 | 0.65 |
| *Cinnamodendron dinisii* | 2.66 | 4.63 | 0.96 | 5.56 | 1.44 | 0.06 | 0.26 |
| *Casearia sylvestris* | 2.58 | 4.63 | 0.96 | 5.56 | 1.44 | 0.04 | 0.18 |
| *Drimys brasiliensis* | 1.91 | 2.31 | 0.48 | 2.78 | 0.72 | 0.17 | 0.71 |
| *Syagrus romanzoffiana* | 1.90 | 2.31 | 0.48 | 2.78 | 0.72 | 0.17 | 0.70 |
| *Lonchocarpus nitidus* | 1.85 | 4.63 | 0.96 | 2.78 | 0.72 | 0.04 | 0.17 |
| *Ocotea puberula* | 1.60 | 2.31 | 0.48 | 2.78 | 0.72 | 0.10 | 0.40 |
| *Machaerium* spp. | 1.38 | 2.31 | 0.48 | 2.78 | 0.72 | 0.04 | 0.18 |
| *Gymnanthes klotzschiana* | 1.28 | 2.31 | 0.48 | 2.78 | 0.72 | 0.02 | 0.08 |
| *Campomanesia xanthocarpa* | 1.25 | 2.31 | 0.48 | 2.78 | 0.72 | 0.01 | 0.05 |
| *Ilex theezans* | 1.24 | 2.31 | 0.48 | 2.78 | 0.72 | 0.01 | 0.04 |
| *Zanthoxylum rhoifolium* | 1.23 | 2.31 | 0.48 | 2.78 | 0.72 | 0.01 | 0.03 |
| *Eugenia uniflora* | 1.23 | 2.31 | 0.48 | 2.78 | 0.72 | 0.01 | 0.03 |
| *Terminalia australis* | 1.22 | 2.31 | 0.48 | 2.78 | 0.72 | 0.00 | 0.02 |

AD = absolute density; ADo = absolute dominance; RDo = relative dominance (%); RD = relative density (%); AF = absolute frequency (%); RF = relative frequency (%).

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
