# Peer review of "Co-Creating Strategies to Optimize Traditional Silvopastoral Systems through the Management of Native Trees in Caívas in Southern Brazil"

_conservation, doi:10.3390/conservation4010005_

Round 1
Reviewer 1 Report
Comments and Suggestions for Authors
It's a very interesting work because in many forests all over the world the abandonment of agricultural practises implies growing of bush with the evident danger in case of forest fire.
As well in south America as other sites as Europe is a very interesting option the cattle management to control the forest fire risk. In this sense I think that would be neccesary a mayor comparative with silvopasture scenarios out of Brazil (USA, Canada, Portugal, Spain, Portugal, France,...).
Author Response
Thank you very much for taking the time to review this manuscript. Your approach to the issue of forest fires is very interesting. This issue has not been a problem in the forest remnants of southern Brazil, due to several factors, especially our rainfall regime (it rains more than 1800mm in the region annually and we do not have a dry period). However, his suggestion caught our attention, because it is yet another issue to be valued in this type of research work.
Our goal is to manage the trees to maintain adequate light levels for pasture growth, which in turn, will maintain well-fed herds within these traditional silvopastoral systems, the caivas. This arrangement will certainly also contribute to healthier landscapes, without excess dry matter, which could potentially cause forest fires.
We included this topic in the discussion, citing some research works that link grazing with forest fires in the United States and Europe. They will certainly contribute to enriching our work. Best regards.
Reviewer 2 Report
Comments and Suggestions for Authors
This is an excellent and detailed study of impacts on species richness and diversity under an improved forest and tree management in the Caivás system.
The study would be considerably strengthened if more information could be provided of overall costs of the forest and pasture management proposed and of the likely benefits both financial and through a stream of environmental services. As the authors acknowledge these economic factors will be critical in the adoption of the proposed methodology in the targeted landscape. This would need some more consideration of the whole farm enterprise and the value generated. It would be very helpful to provide more information about pastures, livestock and yerba mate in the silvopastoral system. Some additional information about the pasture improvement strategy initially introduced would help. Was pasture and dairy cow productivity impacted by the subsequent forest management regime. What sort of annual sales does dairy generate? Also value generated by yerba mate. Any indication of how these changed under forest management and thinning? At the very least we should have an indication that there were no perceived negative economic impacts with the introduction of the forest management methodology.
Other suggestions relating to the line number in the document:
Line 44 “average of 40 species” Per remnant or per hectare?
Line 61 Indicate cut offs by shade level for all categioies of caivas
67 “adoption of technologies proposed by Epagri has led to an increase of up to 400% in pasture production”, how was this measured?
79 Rather than “delicate” it means revising costs and benefits of whole operation.
128 Explain “fall”.
148-166 Management methodology should indicate value of species conserved, what species provide some ecological service eg nitrogen fixation or other benefit of harvestable product. Conserving species on endangered list is environmental service how could this be “valued” or transalated into actual financial flow?
Line 240 Indicate amounts of lime and fertilizer applied.
263 Overall finding of increase in species diversity and richnesss between 2013 and 2020 under improved pasture management very positive.
Line 274 Define SSP
Line 316 should read “abundance “ distribution
Line 345 VI how is importance value defined. What other species had a defined and valued use and give some indication of this.
Figure 7 suggest to present the mean values of PAR as well as median, as the median does not change after tree layer management
Line 399 What is concealer?
Line 407 Economic vision importance in landscape use is mentioned. The author should give this some more consideration ideally with more information on costs and beneftis of the tree management proposed.
417 Can the authors make recommendations for changing the incentive structures to promote good forest management, eg allowing sales of some of harvested wood or through payments for environmental services. What might be the type and value of these environmental services
Comments on the Quality of English Language
Minor editing to improve style is required and a few other small corrections.
Line 36 Correct english. singular
Line 99 Rather than “debt” suggest "obligation"
Use acronyms from english throughout eg Line 366 use PAR (photosynthetically active radiation) rather than RFA
Line 441 “productive fells”, what is this?
Author Response
The authors are grateful for their collaboration, whose suggestions helped to improve the text and correct flaws that had gone unnoticed. We tried to answer all of the reviewer's questions, which are described in the attached file. The remaining requests were all met directly in the final text of the article. Thank you very much for your time and dedication.

Reviewer 3 Report
Comments and Suggestions for Authors
Dear authors,
A well organized and presented empirical analysis on a mixed use forest in Southern Brazil.
I have no major concerns, but minor suggestions:
I would recommend simplifying the first sentence of your abstract.
The second sentence may be a SAF rather than an SAF
44: describe average and standard deviation of tree stands varying from 220 to 1300
47: unquestionable is a strong unjustified claim
76: typo
200: what coordinate system?
Table 1: define or refer readers to what Shannon's diversity index and Pielou's eveness is.
265: typo
287: typo
Consider reviewing tables and figures for appropriate language use.
Well done!
Author Response
The authors are grateful for your collaboration, whose suggestions helped to improve the text and correct flaws that had gone unnoticed. We tried to answer all of the reviewer's questions, which are described in the attached file. The remaining requests were all met directly in the final text of the article. Thank you very much for your time and dedication.

Reviewer 4 Report
Comments and Suggestions for Authors
In my opinion, the present study, which applies sivicultural measures to make the biodiversity of the caíva of southern Brazil compatible with silvopastoral production, is interesting. However, the study, as presented, raises many doubts about the methodology used. It is not clear what silvicultural measures were carried out in the area between 2013 and 2020. The two years evaluated seem to be a short time to observe significant changes. Although the parameters evaluated almost exclusively measure the biodiversity of the site, other degradation parameters are not taken into account, e.g. at the soil level, due to the decrease in coverage. The flora assessed seems to correspond to shrubs and trees, the impact on the flora is not clear. The climatology of the area is also an important factor, since by eliminating the mass of trees, the incidence of the sun is greater, and this depends on the climatology of the place, which is not detailed, nor is it compared with other studies in this sense. There is talk of "epagri" technology, but it is not clear exactly what this is, and the measures applied are very general, without knowing exactly the initial situation of the site and the subsequent result obtained by applying the said measures. In this way, I believe that the results obtained are difficult to interpret, as the results are partly methodologically explained and mainly based on biodiversity indices. At a statistical level, it is not possible to know exactly whether there were significant differences, as they are not shown in the results. For example, the PAR level is only evaluated two years after the treatments have been carried out, i.e. there is no baseline control. If possible, climatic and soil parameters should be correlated with the treatments carried out. At the experimental level, there is no indication of how sampling was carried out, number of samples, replicates, etc. In other words, I think that the manuscript should be significantly improved both at the level of writing, with a reasoned justification based on the bibliography, and at the level of methodology and results. The conclusions are not clear with the results shown and a control and treatments need to be shown and compared statistically. A separate discussion of the results would be desirable and, as I have indicated, more background. The format should also be considered according to the publication rules. The rest of the comments are listed in the attached document.

Comments on the Quality of English LanguageIn my opinion, the manuscript should be more concise, although it needs some minor English editing.
Author Response
Based on your considerations and suggestions in the text, we sought to improve the original version, accepting the majority of the reviewers' opinions, which certainly contributed to the general improvement of the article. We remain at your disposal for any clarifications.

Round 2
Reviewer 4 Report
Comments and Suggestions for Authors
After a second review, I have found that certain improvements have been made and some of the information suggested in the first review has been included. However, I feel that justification should be given as to why the information is not included. In my opinion, the manuscript remains incomplete, with many unresolved methodological issues. The results and discussion sections are basically the same, with a slightly improved background. For my part, I cannot validate this research for publication. Although the editor, if based on other reviewers, feels that it can be published in this form, it is at his discretion.
Comments on the Quality of English LanguageIn my opinion, only minor editing of English language is required.
Author Response
Below is a new text in which we seek to clarify the doubts raised in your comment regarding the article. We also included small adjustments to the text (in green font) to facilitate the understanding of some facts that were not very clear and we removed part of a paragraph from the Introduction, to reduce it.
We hope that we have met your expectations.
Yours sincerely

Round 3
Reviewer 4 Report
Comments and Suggestions for Authors
In my opinion, although the version submitted in this third revision is essentially the same as in the second, I feel that the issues I raised have at least been sufficiently justified. Although I think that the work could be improved, I understand the stated limitations and I think that the work is sufficiently defined. Based on the comments of the other referees, I consider that the work is publishable at the editor's discretion.
Comments on the Quality of English LanguageI consider that only minor editing of the English language is required.
Author Response
We deeply appreciate your commitment, time and valuable contributions to our article. We are very happy with the publication of this work, which, despite its limitations, will certainly contribute to the debate on the use and management of agroforestry systems in Brazil, a topic that still requires a lot of research.
The text was reviewed by an English specialist and we believe that all necessary corrections were made.
Yours sincerely